# Long-term anti-SARS-CoV-2 antibody trajectories after neutralizing monoclonal antibody treatment

Elizabeth S. Munroe[ID][1], Greg A. Grandits[2], Robert C. Hyzy[1], Hallie C. Prescott[1,3], Thomas W. Barrett[4], Robin L. Dewar[5], Nicole Engen[2], Anna L. Goodman[ID][6,7], Timothy J. Hatlen[ID][8], Helene Highbarger[5], Thomas L. Holland[ID][9,10], Gareth Hughes[6], Tomas O. Jensen[11], Muhammad A. Khan[5], Ioannis Kalomenidis[12], Nayon Kang[13], Sylvain Laverdure[14], Prasad Manian[15], Vidya Menon[16], Ravi Patel[17], Srikanth Ramachandruni[18], Tauseef Rehman[19], Kathryn Shaw-Saliba[13], Birgit Thorup Røge[20], David M. Vock[2], Amy C. Weintrob[21], Barnaby E. Young[22], Anne P. Frosch[23]*, for the STRIVE Network and Therapeutics for Inpatients with COVID-19 (TICO) study groups[¶]

1 Division of Pulmonary and Critical Care Medicine, Department of Medicine, University of Michigan, Ann Arbor, Michigan, United States of America, 2 Division of Biostatistics and Health Data Science, University of Minnesota, Minneapolis, Minnesota, United States of America, 3 VA Center for Clinical Management Research, Ann Arbor, Michigan, United States of America, 4 VA Portland Health Care System, and the Department of Medicine, Oregon Health & Science University, Portland, Oregon, United States of America, 5 Frederick National Laboratory for Cancer Research, Frederick, Maryland, United States of America, 6 MRC Clinical Trials Unit, University College London, London, United Kingdom, 7 Centre for Clinical Infection and Diagnostics Research at King's College London and Guy's and St Thomas' NHS Foundation Trust, London, United Kingdom, 8 Division of HIV Medicine, Department of Medicine, Harbor-UCLA Medical Center, Torrance, California, United States of America, 9 Department of Medicine, Division of Infectious Diseases, Duke University, Durham, North Carolina, United States of America, 10 Duke Clinical Research Institute, Durham, North Carolina, United States of America, 11 Centre of Excellence for Health, Immunity, and Infections, Rigshospitalet, University of Copenhagen, Denmark, 12 1st Department of Critical Care and Pulmonary Medicine, National and Kapodistrian University of Athens, School of Medicine, Evaggelismos Hospital, Athens, Greece, 13 Division of Clinical Research, National Institute of Allergy and Infectious Diseases, Bethesda, Maryland, United States of America, 14 Laboratory of Human Retrovirology and Immunoinformatics, Frederick National Laboratory, Frederick, Maryland, United States of America, 15 Division of Pulmonary and Critical Care Medicine, Department of Medicine, Baylor College of Medicine, Houston, Texas, United States of America, 16 Department of Medicine, New York City Health and Hospitals, Bronx, New York, United States of America, 17 V.A. Bay Pines Health Care System, Florida, United States, 18 Christus Spohn Hospital, Corpus Christi, Texas, United States of America, 19 Leidos Biomedical Research Inc., Frederick, Maryland, United States of America, 20 Department of Internal Medicine, Lillebælt Hospital, Kolding, Denmark, 21 Infectious Diseases Section, Washington Veterans Affairs Medical Center, Washington, DC, United States of America, 22 National Centre for Infectious Diseases, Singapore, 23 Department of Medicine, Hennepin Healthcare Research Institute, University of Minnesota, Minneapolis, Minnesota, United States of America

¶ Membership of the STRIVE Network and Therapeutics for Inpatients with COVID-19 (TICO) study Groups is provided in the Acknowledgements
* Anne.frosch@hcmed.org

## Abstract

### Background

Neutralizing monoclonal antibodies (nMAbs) have been used to treat COVID-19 and are increasingly being used to treat other infections. However, there is concern that by neutralizing the SARS-CoV-2 virus, nMAbs may decrease the availability of

---

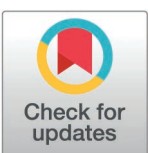

**Data availability statement:** Data for the ACTIV-3/TICO trials are publicly available at https://public-data.ccbr.umn.edu/.

**Funding:** This research was, in part, funded by the National Institutes of Health (NIH) Agreement 1OT2HL156812-01. The trial was sponsored and primarily funded by the National Institute of Allergy and Infectious Diseases (NIAID), National Institutes of Health (NIH), Bethesda, MD, in part with federal funds from the NIAID and the National Cancer Institute, NIH, under contract 75N91019D00024, task order numbers 75N91020F00014 and 75N91020F00039. The work was funded under Subcontract 18X107C under Leidos Biomeds's Prime Contract HHSN261200800001E, NIH. NIH Grant U01-AI136780. The following authors received relevant funding support: • Author ESM was supported by Grant Number F32 HL 172463 from the National Institutes of Health, National Heart, Lung, and Blood institute. • Author BEY was supported by Singapore National Medical Research Council (NMRC, grant number COVID19RF-0005). • Author ALG receives funding to support salary from the Medical Research Council (MC_UU_00004/05).

**Competing interests:** Author BEY has received honoraria/speaker fees from Astra-Zeneca, Gilead, Sanofi, Pfizer, Moderna (paid to their institution and outside of the submitted work). All other authors report no competing interests.

antigens to the immune system, potentially impairing the endogenous polyclonal immune response and decreasing long-term immune protection.

## Methods

We compared 28 and 90-day anti-SARS-CoV-2 spike protein neutralization activity and anti-SARS-CoV-2 nucleocapsid response for patients hospitalized with COVID-19 infection randomized to receive nMAbs or placebo in the large platform ACTIV-3/TICO trials. We pooled results from four trials of anti-spike nMAbs. For most tested agents, measurements of the spike protein response reflect both the therapeutic and endogenous immune response. Anti-nucleocapsid levels reflect only the endogenous immune response. Data are summarized as mean differences in percent binding inhibition (anti-spike) and signal-to-cutoff (S/C) ratio (anti-nucleocapsid). Linear mixed effects models were fit to compare the longitudinal trajectory between treatment and placebo groups.

## Results

Of 2,254 participants in the ACTIV-3/TICO trials modified intention-to-treat population, 2,149 (95.3%) had antibody measures at baseline and at least 1 follow-up day (day 1, 3, or 5) and were included in this analysis. Antibody measures were available for 1,556 (72.4%) participants at day 28 and 1,429 (66.5%) participants at day 90. In participants who received nMAbs, anti-spike neutralization activity was higher at day 28 (mean difference in percent binding inhibition: 7.1% [95%CI: 5.3, 8.9], $p < 0.001$) and day 90 (mean difference in percent binding inhibition: 7.2% [95% CI: 5.4, 9.0], $p < 0.001$). Anti-nucleocapsid response was similar at day 28 (mean difference in S/C ratio: 0.02 [95%CI: −0.11, 0.15], $p = 0.75$) and day 90 (mean difference in S/C ratio: 0.08 [95% CI: −0.05, 0.21], $p = 0.22$). Similar patterns were observed in all trials.

## Conclusions

In patients hospitalized with COVID-19, treatment with nMAbs did not decrease long-term anti-nucleocapsid response compared to placebo, suggesting neutralizing therapies do not suppress the endogenous humoral immune response in this population.

## Introduction

Passive immunization with neutralizing monoclonal antibodies (nMAbs) has been successful in managing several viral infections, including preventing RSV in infants and treating Ebola [1,2]. NMAbs also played an important role in COVID-19 primary prevention and post-exposure prophylaxis [3–5]. Additionally, several large, multi-center randomized controlled trials found that administering nMAbs against the SARS-CoV-2 spike protein reduced COVID-19-related hospitalization and death in outpatients with mild-to-moderate COVID-19 infection and risk factors for disease progression [6–8]. However, nMAbs have been less successful in treating patients hospitalized with

COVID-19, with only one trial suggesting a potential mortality benefit and all finding no difference in 90-day sustained recovery [9–11]. As the SARS-CoV-2 virus has evolved, nMAbs designed against previous variants have also lost efficacy [12–17].

Despite the shortcomings of nMAbs in treating severe COVID-19 infection, passive immunization remains an important tool for treating novel viral infections given the feasibility of rapid development and deployment of treatments like nMAbs and targeted small molecules. Future use of such therapies should be undertaken with a full understanding of potential risks. It has been proposed that neutralizing treatments may impair long-term B cell responses by limiting antigen availability during the immune response [18]. In SARS-CoV-2 for example, neutralizing the virus with nMAbs targeted to the spike protein could decrease viral antigen presentation to B cells or alter innate immune signaling and thus change the host response to the virus. A variety of surrogates for the endogenous immune response have been used to evaluate the effect of nMAb treatment on the host response, including anti-spike IgM, anti-nucleocapsid response, and specific antibody neutralizing activities [18,19]. Previous small studies have demonstrated decreases in multiple of these measures with anti-SARS-CoV-2 nMAb treatments [18–21]. This has led to the concern that use of antigen-specific neutralizing therapies may alter the host immune response to natural infection, attenuating the intensity of the adaptive humoral immune response and resulting in less robust long-term protection and impaired vaccine immunity [22].

The Accelerating COVID-19 Therapeutic Interventions and Vaccines–Therapeutics for Inpatients with COVID-19 platform (ACTIV-3/TICO) was a large trial platform that followed participants hospitalized with COVID-19 through early convalescence. The platform design allows the effect of nMAbs on endogenous immunity to be examined across several large trials. We assessed 28-day and 90-day antibody responses among participants hospitalized for COVID-19 in four placebo controlled trials of nMAbs [9–11]. The aim of the study was to compare 28-day and 90-day antibody responses with and without nMAb treatment. Given that the SARS-CoV-2 nucleocapsid protein was not a target of any ACTIV-3/TICO therapeutics, we used nucleocapsid antibodies as a surrogate for the endogenous host humoral immune response consistent with prior studies [19,21,23]. In addition, we evaluated the antibody response from two other ACTIV-3/TICO placebo-controlled trials: 1) a neutralizing anti-spike molecule and 2) a small molecule viral proteinase [24], which allow evaluation of host response in the presence of targeted therapies on the same trial platform.

## Methods

### ACTIV-3/TICO platform

This was a pre-specified secondary analysis of trials conducted through the ACTIV-3/TICO platform (NCT04501978), a phase III multicenter, adaptive, randomized, blinded platform trial of therapeutics for hospitalized patients with COVID-19 [25]. In this study, we compared long-term (28-day and 90-day) antibody responses among patients treated with nMAb vs placebo across four trials of novel SARS-CoV-2 nMAbs. We also compared long-term antibody responses among patients treated with two other molecules with activity against SARS-CoV-2 vs placebo. Patients were enrolled in the ACTIV-3/TICO platform trials between August 5, 2020 and November 15, 2021 and followed for a minimum of 90 days. Written informed consent was obtained from all participants or their legally authorized representatives. As part of a shared platform, all trials included similar eligibility criteria, enrolling patients ≥18 years old hospitalized for laboratory-confirmed SARS-CoV-2 infection who had COVID-19 symptoms for ≤12 days. All trials enrolled hospitalized patients with moderate to severe COVID-19 infection without organ failure or major extrapulmonary involvement. Full eligibility criteria are described in detail in the original trials [9–11,26]. The studies were approved by a governing institutional review board for each enrolling site.

### ACTIV-3/TICO treatments

The ACTIV-3/TICO trials evaluated four nMAb/ nMAb combinations directed against the SARS-CoV-2 spike protein: Bamlanivimab (LY-CoV555, Eli Lilly and Company), Sotrovimab (VIR-7831, Vir biotechnology and GlaxoSmithKline), Amubarvimab/Romlusevimab (BRII-196 and BRII-198, BRII Biosciences), and Tixagevimab/Cilgavimab (AZD7442,

AstraZeneca). These medications were given as one-time infusions. Due to modifications in the Fc chain, these nMAb treatments have varying half-lives, ranging from 20 days for Bamlanivimab to 80−90 days for Amubarvimab/Romluse-vimab and Tixagevimab/Cilgavimab (Table 1) [27–31]. In addition, we examined antibody responses after treatment with two non-antibody molecules evaluated on the ACTIV-3/TICO platform: 1) ensovibep (MP), a designed ankyrin repeat protein (DARPin) that targets and neutralizes the SARS-CoV-2 spike protein [32] and 2) lufotrelvir (PF-07304814), a phosphate ester pro-drug that is a selective inhibitor of the SARS-CoV-2 3CLpro, a 3C-like main protease [33]. All agents with the exception of Lufotrelvir target the SARS-CoV-2 spike protein. We included Lufotrelvir because it was included in the pre-specified plan for this secondary analysis, which aimed to evaluate long-term antibody responses to all therapies tested on the ACTIV-3/TICO platform.

**Table 1. Overview of the six original ACTIV-3/TICO trials.**

| Agent | Abbr. Name | Drug Name [Half Life] | Biological Name | Mechanism of Action | Intv. Arm N* | Placebo Arm N* | Original trial dates | # of sites | Primary Outcome** | Status |
|---|---|---|---|---|---|---|---|---|---|---|
| A(9) | LILLY | Bamlanivimab [20.9 days] (46) | LY-CoV55, or LY3819253 | nMAb | 163 | 151 | 8/2020-10/2020 | 31 | Sustained recovery†: 82% vs 79%, RR: 1.06 [95%CI: 0.77–1.47] | Stopped early for futility |
| B(10) | VIR | Sotrovimab [61 days] (47–50) | VIR-7831 | nMAb *(Modified Fc for extended half-life)* | 182 | 178‡ | 12/2020− 3/2021 | 43 | Sustained recovery†: 88% vs 85%, aRR: 1.12 [95%CI: 0.91–1.37] | Stopped early for futility |
| C(10) | BRII | Amubarvimab/ Romlusevimab [90 days] (10) | BRII-196 + BRII-198 | nMAb *(Modified to extend half-life, minimize ADEs)* | 176 | 178‡ | 12/2020− 3/2021 | 43 | Sustained recovery†: 88% vs 85%, aRR 1.08 [95% CI: 0.88–1.32] | Stopped early for futility |
| D(11) | AZ | Tixagevimab/ Cilgavimab [87.9 days/ 82.9 days] (37,38) | AZD7442 | nMAb *(Modified to extend half-life, reduce Fc effector function)* | 710 | 707 | 2/2021-9/2021 | 81 | Sustained recovery†: 89% vs 86%, RRR 1.08 [95%CI: 0.97–1.20] Mortality was lower in treatment group: 9% vs 12%, HR = 0.70 [95%CI: 0.5–0.97], p = 0.032 | Full study completed |
| E(26) | MP | ensovibep [13 days] (31) | MP0420 | Small molecule: Designed ankyrin repeat proteins (DARPins) | 247 | 238 | 6/2021-11/2021 | 62 | Sustained recovery†: 82% vs 80%, sHR: 1.06 [95% CI: 0.88–1.28]) | Stopped early for futility |
| F(24) | PF | Lufotrelvir [2 hours] (51) | PF-07304814 | Small molecule: inhibitor of SARS-CoV-2 3CLpro | 32 | 26 | 9/2021- *sus-pended by FDA as of 3/2023* | 11 | n/a | Suspended. Agent with-drawn from development by sponsor |

Grey boxes highlight the primary analysis, pooling neutralizing monoclonal antibodies

†Sustained recovery was defined as discharge home with at least 14 consecutive days at home, up to day 90 after randomization.

‡Sotrovimab and Amubarvimab/ Romlusevimab were assessed in the same trial and share a placebo group.

*Included in treatment by modified intention to treat (mITT) if they received any amount of the treatment agent

**Presented as intervention vs placebo.

Definitions: Abbr. = Abbreviated, Intv = Intervention, nMAb = neutralizing monoclonal antibody, RR = rate ratio, CI = confidence interval, Fc = fragment crystallizable region, aRR = adjusted rate ratio, RRR = recovery rate ratio, HR = hazard ratio, sHR = subhazard ratio, Fc = fragment crystallizable antibody region, ADE = adverse drug effects

## Study population and treatment assignment

In the ACTIV-3/TICO trials, participants were randomized to active treatment vs placebo. In some cases a placebo participant was used as a control for multiple trials [9–11,26]. Due to shared placebos, more participants were randomized to active treatment than placebo. Participants who received any amount of treatment or placebo were included in the modified intention to treat (mITT) population. Participants in the mITT population who had antibody levels measured at baseline and at least one follow-up day (1, 3 or 5) were included in this secondary analysis, consistent with earlier analyses [34].

## SARS-CoV2 antigen and antibody quantification

Per the ACTIV-3/TICO master protocol, antigen and antibody (anti-spike and anti-nucleocapsid) levels were measured at enrollment (day 0) and days 1, 3, 5, 28, and 90. Measurements were performed centrally on stored plasma specimens using the following assays:

**Anti-SARS-CoV-2 spike neutralization.** GenScript SARS-CoV-2 Surrogate Virus Neutralization Test (sVNT) was used to estimate neutralization directed against the SARS-CoV-2 receptor binding domain (RBD) on the spike protein and the ACE2 receptor (GenScript, Piscataway, New Jersey) [35,36]. Results are expressed as percent binding inhibition, with >30% considered positive. Most of the studied nMAbs and ensovibep bind to this RBD and are detected by this assay [37,38], with the exception of Sotrovimab. Sotrovimab blocks viral fusion by recognizing a proteoglycan epitope distinct from the RBD and therefore has lower detection by this assay [38].

**Anti-SARS-CoV-2 nucleocapsid antibodies.** BioRad Platelia SARS-CoV-2 Total Ab assay was used to measure total immunoglobulin (IgG, IgA and IgM) against the SARS-CoV-2 nucleocapsid antigen (BioRad, Hercules, California). This assay uses a one-step antigen capture format enzyme-linked immunosorbent assay (ELISA) [39]. Results are reported as signal-to-cutoff ratios (S/C ratios), which are defined as specimen optical density (OD) divided by control OD [control $R4(OD_MR4)$]. S/C ratios ≥ 1.0 are considered positive.

**N protein antigen levels.** SARS-CoV-2 nucleocapsid antigen levels were measured using the Quanterix assay (Quanterix, Billerica, MA). This assay is an automated paramagnetic 2-step microbead-based sandwich ELISA [40]. Antigen results are reported as concentrations in pg/ml. The lower level of quantification was 3 pg/ml. Results < 3 pg/ml were imputed as 2.9 pg/ml.

## Outcomes

Outcomes were pre-specified in the ACTIV-3/TICO trial design and included: anti-spike neutralization activity, anti-nucleocapsid response, and SARS-CoV-2 antigen levels at days 1, 3, 5, 28, and 90. This manuscript focuses on long-term outcomes (day 28 and 90); data on short-term (day 1–5) antibody responses have been previously published [34].

## Statistical analysis

This study was a pre-specified secondary analysis of trials performed on the ACTIV-3/TICO platform. The primary goal was to compare outcomes between pooled nMAb treatment and placebo groups. We also compared outcomes for each studied nMAb/ nMAb combination (4 trials), ensovibep, and lufotrelvir individually. Longitudinal plots of means with standard errors at days 0, 1, 3, 5, 28, and 90 are presented by treatment versus placebo. Mean differences in antibody levels between treatment and placebo groups at each follow-up day were estimated using longitudinal linear mixed effects models. Participants randomized to a placebo group that was shared across trials were weighted by the number of agents for which they served as controls to account for varying randomization ratios (treatment/placebo) over time. Day (as a categorical factor), treatment group, day by treatment group interaction, and baseline antibody activity (anti-spike) or level (anti-nucleocapsid) were included as fixed effects in the model. A random intercept was included for each participant to account for correlation in values over time. For antigen analysis, we obtained maximum likelihood estimates of the

parameters in the mixed model to account for left censoring of antigen values below the limit of quantification [41,42]. Antigen levels were log-transformed for analyses, and results were back-transformed to the original scale (geometric means). To account for loss to follow-up due to death and missing blood draws, we conducted a sensitivity analysis restricted to participants in the four nMAb trials who had antibody levels at all time periods.

In an exploratory epidemiologic analysis, similar longitudinal models were used to compare anti-nucleocapsid antibody response across groups of participants defined *a-priori* based on the following factors: baseline anti-spike positivity, age (≥65 years old versus < 65 years), sex, body mass index (≥30 kg/m$^2$ versus < 30 kg/m$^2$), diabetes mellitus, kidney disease, immunosuppression (defined as pre-existing immunocompromising condition, active malignancy, or treatment with immunosuppressive medications), oxygen status (≥4 Liters per minute versus < 4 Liters per minute), and vaccination status (fully vaccinated versus partial or no vaccination with full vaccination defined based on Centers for Disease Control and Prevention (CDC) vaccine recommendations at the time). Active and placebo groups across all ACTIV-3/TICO trials were combined in these exploratory analyses, given there was limited power to examine long-term antibody responses by both baseline characteristic and treatment assignment.

P-values <0.05 were considered significant. No adjustments were made for multiple testing.

Data analysis was conducted in SAS Version 9.4 (SAS Institute, Cary, NC).

## Results

Of 2,313 participants enrolled in ACTIV-3/TICO trials of nMAbs [Bamlanivimab, Sotrovimab, Amubarvimab/Romlusevimab, Tixagevimab/Cilgavimab], 2,254 received all or part of the treatment infusion or placebo and were included in the mITT population. Of participants in the mITT population, 2,149 (95.3%) had baseline and ≥1 follow-up (day 1, 3, or 5) antibody measures and were included in this study, including 1,178/1,231 (95.2%) participants in the treatment groups and 971/1,023 (94.9%) participants in the placebo groups [Fig 1]. All available antibody measures per participant were used. In the treatment groups, plasma specimens were available for antibody assays at day 28 in 859/1,120 participants alive at day 28 (76.7%) and in 791/1,082 participants alive at day 90 (73.1%). In the placebo groups, plasma specimens were available for antibody level measurement at day 28 in 697/905 participants alive at day 28 (77.0%) and day 90 in 638/878 participants alive at day 90 (72.7%). See study flow diagram in Fig 1 for details.

Overall median age of participants was 57 years old; 58% were male, 83.4% had at least one significant comorbidity and/or BMI ≥ 30 kg/m$^2$, 82% were unvaccinated, 80% were enrolled from North America, and 74% were receiving supplemental oxygen at enrollment [Table 2]. A majority of participants had the ancestral SARS-CoV-2 variant (1474, 69.1%). The remaining 658 (30.9%) participants had the Delta variant. All participants with the Delta variant were enrolled in the later Tixagevimab/Cilgavimab trial.

Compared to placebo, anti-spike neutralization activity rose faster in participants who received nMAbs (treatment-placebo group differences by day, p < 0.001), reflecting treatment with the anti-spike nMAb. Anti-spike neutralization activity remained higher in the treated groups at day 28 (mean difference in percent binding inhibition: 7.1% [95%CI: 5.3, 8.9], p < 0.001) and day 90 (mean difference in percent binding inhibition: 7.2% [95% CI: 5.4, 9.0], p < 0.001) [Table 3; Fig 2A]. The anti-spike neutralization activity had a similar trajectory compared to placebo for all nMAbs with the exception of Sotrovimab, which has a primary target outside of the RBD [38]. Sotrovimab had smaller increases in anti-spike compared to the other nMAb agents combined (p < 0.001) (S2 Fig).

The anti-nucleocapsid response was similar between nMAb and placebo groups at day 28 (mean difference in S/C ratio: 0.02 [95%CI: −0.11, 0.15], p = 0.75) and at day 90 (mean difference in S/C ratio: 0.08 [95% CI: −0.05, 0.21], p = 0.22) [Table 3, Fig 2B].

As reported previously, participants who received nMAbs had lower antigen levels at day 3 (geometric mean ratio 0.64 [95% CI: 0.56, 0.74], p < 0.001) and day 5 (geometric mean ratio: 0.62 [95% CI: 0.54, 0.72], p < 0.001) [34] [Table 3, | Fig 2C]. Antigen levels continued to decline in both groups, and by day 28, antigen levels were similar between treatment

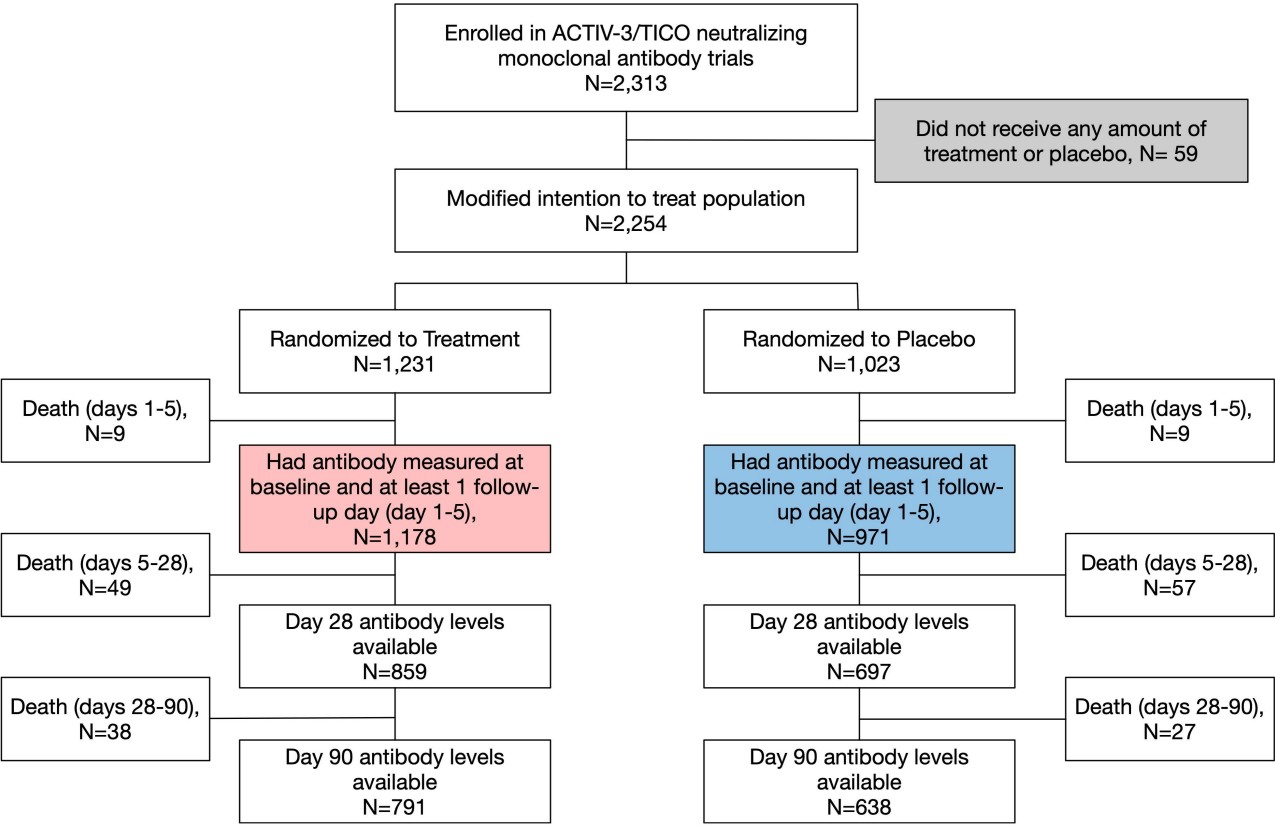

**Fig 1. Study Flow Diagram.** Study flow diagram for the pooled analysis. There were 96 deaths in the active groups and 93 deaths in the placebo groups through Day 90. Red box = participants included in the treatment groups. Blue box = participants included in the placebo groups. Of note, in some cases a placebo participant was used as a control for multiple trials [9–11,26]. Due to shared placebos, more participants were randomized to active treatment than placebo.

and placebo groups, with median and interquartile range levels below the detectable cut-off (2.9 pg/mL). Both groups had a small number of patients with antigen positivity at day 28 (1.3% in treatment vs 1.1% in placebo) and day 90 (0.3% in treatment vs. 0.6% in placebo) [Table 3].

Trajectories and levels of anti-nucleocapsid response were similar for individual nMAb agents, ensovibep, and lufotrelvir (S1 – S6 Figs). Similar results were seen in sensitivity analysis restricting the study population to patients in the pooled nMAb trials who were alive and had antibody/antigen levels measured at all time points through day 90 (S7 Fig. and S1 Table).

### Epidemiologic analysis

In epidemiologic analysis, results from all six trials were pooled. Participants who were positive for anti-spike neutralization activity at baseline (>30% binding inhibition) had higher anti-nucleocapsid response through day 90 than participants negative for anti-spike at baseline, regardless of treatment group (S8 Fig). Anti-nucleocapsid response was initially lower and remained lower at days 28 and 90 for participants who were immunosuppressed, fully vaccinated, and on <4 L oxygen at baseline (S9 Fig). Long-term anti-nucleocapsid response did not differ or differed only slightly across time points by BMI, sex, or history of diabetes mellitus. Participants ≥65 years old or with a history of renal failure had lower anti-nucleocapsid response on days 1–5, but response was similar by day 28 (S9 Fig).

**Table 2. Baseline patient characteristics across neutralizing monoclonal antibody trials.**

| | Pooled | Bamlanivimab | Sotrovimab | Amubarvimab/ Romlusevimab | Tixagevimab/ Cilgavimab |
|---|---|---|---|---|---|
| No. participants; total (treatment) | 2149 (1178) | 306 (159) | 254 (172) | 250 (167) | 1339 (680) |
| **Age**, years median (IQR) | 57 (46, 68) | 61 (49, 71) | 60 (50, 72) | 60 (49, 71) | 54 (44, 66) |
| **Sex**, Male, N(%) | 1246 (58.0) | 174 (56.9) | 150 (59.1) | 143 (57.2) | 779 (58.2) |
| **BMI**, kg/m$^2$ median (IQR) | 31 (26,36) | 30 (26,36) | 31 (27,37) | 30 (27,35) | 31 (26,36) |
| BMI ≥ 30 kg/m$^2$ | 1151 (53.7) | 161 (52.8) | 141 (55.5) | 129 (51.6) | 720 (54.0) |
| BMI ≥ 40 kg/m$^2$ | 307 (14.3) | 39 (12.8) | 42 (16.5) | 31 (12.4) | 195 (14.6) |
| **Comorbidities**, N(%) | | | | | |
| Cardiovascular disease | 1033 (48.1) | 165 (53.9) | 147 (57.9) | 148 (59.2) | 573 (42.8) |
| Chronic kidney disease | 211 (9.8) | 32 (10.5) | 37 (14.6) | 19 (7.6) | 123 (9.2) |
| Chronic lung disease | 326 (15.2) | 44 (14.4) | 40 (15.7) | 44 (17.6) | 198 (14.8) |
| Diabetes | 618 (28.8) | 89 (29.1) | 98 (38.6) | 87 (34.8) | 344 (25.7) |
| Hepatic impairment | 36 (1.7) | 1 (0.3) | 5 (2.0) | 6 (2.4) | 24 (1.8) |
| HIV | 36 (1.7) | 2 (0.7) | 5 (2.0) | 2 (0.8) | 27 (2.0) |
| Immunocompromised* | 328 (15.3) | 29 (9.5) | 33 (13.0) | 36 (14.4) | 230 (17.2) |
| Any of the above or BMI ≥ 30 kg/m$^2$ | 1792 (83.4) | 256 (83.7) | 228 (89.8) | 223 (89.2) | 1085 (81.0) |
| **Geographic Region**, N(%) | | | | | |
| Africa | 86 (4.0) | 0 (0.0) | 0 (0.0) | 0 (0.0) | 86 (6.4) |
| Asia | 24 (1.1) | 1 (0.3) | 0 (0.0) | 0 (0.0) | 23 (1.7) |
| Europe | 325 (15.1) | 37 (12.1) | 16 (6.3) | 12 (4.8) | 260 (19.4) |
| North America | 1714 (79.8) | 268 (87.6) | 238 (93.7) | 238 (95.2) | 970 (72.4) |
| **Vaccination status****, N(%) | | | | | |
| Fully vaccinated | 190 (8.8) | 0 (0.0) | 1 (0.4) | 0 (0.0) | 189 (14.1) |
| Partially vaccinated | 195 (9.1) | 0 (0.0) | 17 (6.7) | 15 (6.0) | 163 (12.2) |
| Not vaccinated | 1764 (82.1) | 306 (100) | 236 (92.9) | 235 (94.0) | 987 (73.7) |
| **Days since symptom onset**, median (IQR) | 8 (6,10) | 7 (5,9) | 8 (5,9) | 8 (5,9) | 8 (6,10) |
| **Oxygen status, baseline**, N(%) | | | | | |
| No supplemental O2 | 553 (25.7) | 87 (28.4) | 84 (33.1) | 80 (32.0) | 302 (22.6) |
| O2 < 4 L/min | 816 (38.0) | 111 (36.3) | 114 (44.9) | 102 (40.8) | 489 (36.5) |
| O2 ≥ 4 L/min | 582 (27.1) | 62 (20.3) | 56 (22.0) | 68 (27.2) | 396 (29.6) |
| NIV or HFNC | 198 (9.2) | 46 (15.0) | 0 (0.0) | 0 (0.0) | 152 (11.4) |
| **Pre-enrollment medications**, N(%) | | | | | |
| Corticosteroids | 1465 (68.2) | 155 (50.7) | 165 (65.0) | 156 (62.4) | 989 (73.9) |
| Heparin, therapeutic | 85 (4.0) | 6 (2.0) | 6 (2.4) | 4 (1.6) | 69 (5.2) |
| Remdesivir | 1989 (92.6) | 294 (96.1) | 231 (90.9) | 223 (89.2) | 1241 (92.7) |
| **Viral Variant**, N(%) | | | | | |
| Delta | 658 (30.9) | 0 (0.0) | 0 (0.0) | 0 (0.0) | 658 (49.8) |
| Not Delta (Ancestral) | 1474 (69.1) | 306 (100) | 254 (100) | 250 (100) | 664 (50.2) |

Table 2 Legend. Baseline patient characteristics in pooled neutralizing monoclonal antibody (nMAb) trials and by agent. Data are presented as N(%) or median (IQR). Included agents were: Bamlanivimab (LY-CoV555, Eli Lilly and Company), Sotrovimab (VIR-7831, Vir biotechnology and GlaxoSmith-Kline), Amubarvimab/ Romlusevimab (BRII-196 and BRII-198, BRII Biosciences), Tixagevimab/ Cilgavimab (AZD7442, AstraZeneca). IQR = interquartile range, O2 = oxygen, NIV = non-invasive positive pressure ventilation, HFNC = high flow nasal cannula.

*Immunocompromised: pre-existing immunocompromising condition, active malignancy, or treatment with immunosuppressive medications

***Fully vaccinated = full course completed, symptoms started at least 14 days after the last dose; partially vaccinated = full course complete and symptoms started within 14 days after last dose, or 2-dose course and only 1 dose received; Not vaccinated = first dose received after symptoms start, or no doses received/ unknown.

**Table 3. Quantitative antibody and antigen responses to nMAb vs placebo day 1-90: Pooled analysis of neutralizing monoclonal antibodies.**

| | Treatment | | | | Placebo | | | | Estimated Mean Difference/ Geometric Mean Ratio (95%CI)[b] | P-value | P-value (Day x txt) |
|---|---|---|---|---|---|---|---|---|---|---|---|
| | N | N (%) positive[a] | Mean | Median (IQR) | N | N (%) positive[a] | Mean | Median (IQR) | | | |
| **Anti-spike antibody (% binding inhibition)** | | | | | | | | | | | |
| Baseline | 1178 | 595 (50.5%) | 38.28 | 30.5 (7.4 - 68.9) | 970 | 470 (48.5%) | 37.86 | 28.3 (8.9 - 69.8) | n/a[⊥] | n/a[⊥] | <0.001 |
| 1 | 1128 | 1119 (99.2%) | 92.10 | 97.6 (97.0 - 97.8) | 923 | 597 (64.7%) | 49.63 | 49.5 (18.3 - 81.4) | 42.59 (40.98, 44.19) | <0.001 | |
| 3 | 1033 | 1028 (99.5%) | 93.72 | 97.7 (97.1 - 97.9) | 870 | 712 (81.8%) | 65.62 | 77.1 (43.9 - 90.8) | 27.72 (26.07, 29.37) | <0.001 | |
| 5 | 970 | 966 (99.6%) | 94.28 | 97.6 (97.1 - 97.8) | 841 | 760 (90.4%) | 76.75 | 87.1 (72.6 - 93.6) | 17.82 (16.15, 19.50) | <0.001 | |
| 28 | 859 | 848 (98.7%) | 93.85 | 97.0 (96.0 - 98.0) | 696 | 683 (98.1%) | 88.03 | 94.0 (88.0 - 96.0) | 7.09 (5.32, 8.86) | <0.001 | |
| 90 | 791 | 785 (99.2%) | 94.60 | 97.0 (97.0 - 98.0) | 638 | 618 (96.9%) | 88.36 | 96.0 (88.0 - 97.0) | 7.20 (5.38, 9.02) | <0.001 | |
| **Anti-nucleocapsid antibody (S/C ratio)** | | | | | | | | | | | |
| Baseline | 1178 | 709 (60.2%) | 2.35 | 2.8 (0.2 - 4.1) | 971 | 621 (64.0%) | 2.50 | 3.2 (0.2 - 4.1) | n/a[⊥] | n/a[⊥] | 0.41 |
| 1 | 1130 | 783 (69.3%) | 2.74 | 3.6 (0.5 - 4.2) | 925 | 648 (70.1%) | 2.87 | 3.7 (0.5 - 4.2) | −0.05 (−0.16, 0.06) | 0.41 | |
| 3 | 1034 | 869 (84.0%) | 3.36 | 3.9 (3.1 - 4.2) | 870 | 731 (84.0%) | 3.36 | 3.9 (3.1 - 4.2) | 0.04 (−0.07, 0.16) | 0.47 | |
| 5 | 969 | 870 (89.8%) | 3.68 | 4.0 (3.5 - 4.4) | 844 | 779 (92.3%) | 3.77 | 4.1 (3.5 - 4.4) | −0.02 (−0.14, 0.10) | 0.72 | |
| 28 | 859 | 817 (95.1%) | 4.81 | 5.0 (4.7 - 5.3) | 697 | 671 (96.3%) | 4.86 | 5.0 (4.7 - 5.3) | 0.02 (−0.11, 0.15) | 0.75 | |
| 90 | 791 | 735 (92.9%) | 4.54 | 5.0 (4.3 - 5.3) | 638 | 608 (95.3%) | 4.46 | 4.9 (4.2 - 5.1) | 0.08 (−0.05, 0.21) | 0.22 | |
| **SARS-CoV-2 antigen (pg/mL)[c]** | | | | | | | | | | | |
| Baseline | 1178 | 1107 (94.0%) | 847.4 | 1535 (272 - 4678) | 969 | 925 (95.5%) | 844.5 | 1428 (235 - 4710) | n/a[⊥] | n/a[⊥] | <0.001 |
| 1 | 1129 | 1049 (92.9%) | 442.5 | 708.5 (60 - 3546) | 923 | 864 (93.6%) | 474.9 | 696.9 (93 - 3527) | 0.91 (0.80, 1.04) | 0.17 | |
| 3 | 1033 | 849 (82.2%) | 43.4 | 33.0 (6.7 - 156.0) | 869 | 749 (86.2%) | 62.0 | 44.1 (9.9 - 238.2) | 0.64 (0.56, 0.74) | <0.001 | |
| 5 | 970 | 572 (59.0%) | 12.0 | 5.9 (2.9 - 28.7) | 844 | 532 (63.0%) | 15.3 | 7.5 (2.9 - 32.9) | 0.62 (0.54, 0.72) | <0.001 | |
| 28 | 859 | 11 (1.3%) | 4.0 | 2.9 (2.9 - 2.9) | 696 | 8 (1.1%) | 4.0 | 2.9 (2.9 - 2.9) | 0.99 (0.62, 1.60) | 0.98 | |
| 90 | 791 | 2 (0.3%) | 3.9 | 2.9 (2.9 - 2.9) | 637 | 4 (0.6%) | 4.0 | 2.9 (2.9 - 2.9) | 0.48 (0.24, 0.99) | 0.05 | |

Table 3 legend: Raw mean and median antibody and antigen levels from baseline to day 90 in pooled trials of neutralizing monoclonal antibodies within the treatment and placebo group.

[a]Positivity for each assay was determined using the following parameters: Anti-spike neutralization activity (GenScript AB): > 30% binding inhibition, Anti-nucleocapsid levels (BioRad AB): S/C ratio ≥1, SARS-CoV-2 antigen level (Quanterix AG): ≥ 3 pg/mL.

[b]Estimates from longitudinal model comparing active versus placebo at each time point, adjusting for baseline level of blood marker.

[c]SARS-CoV-2 antigen levels were log-transformed for analyses, and results were back-transformed to the original scale. Results of antigen levels are therefore presented as geometric means and geometric mean ratio (95% CI) for comparison.

[⊥]Baseline levels were not compared given the studies were all randomized controlled trials.

*Definitions*: IQR = interquartile range, CI = confidence interval, S/C ratio = signal-to-cutoff ratio: specimen optical density divided by control optical density as measured using the BioRad Assay.

## Discussion

In this prespecified secondary analysis of over 2,000 patients enrolled in ACTIV-3/TICO randomized trials, participants hospitalized for severe COVID-19 who were treated with SARS-CoV-2 anti-spike nMAbs had similar 28-day and 90-day anti-nucleocapsid responses as participants who received placebo. These results suggest that treatment with nMAbs during hospitalization for severe COVID-19 does not impair host humoral immune response to the target antigen or other viral antigens. Similar findings were demonstrated for the neutralizing anti-spike DARPin studied on this platform.

Given nMAbs can be faster to develop than vaccines, they have gained attention for their potential role in combatting novel viral infections, including Ebola and COVID-19 [2–4,6]. There has also been interest in using nMAbs to combat

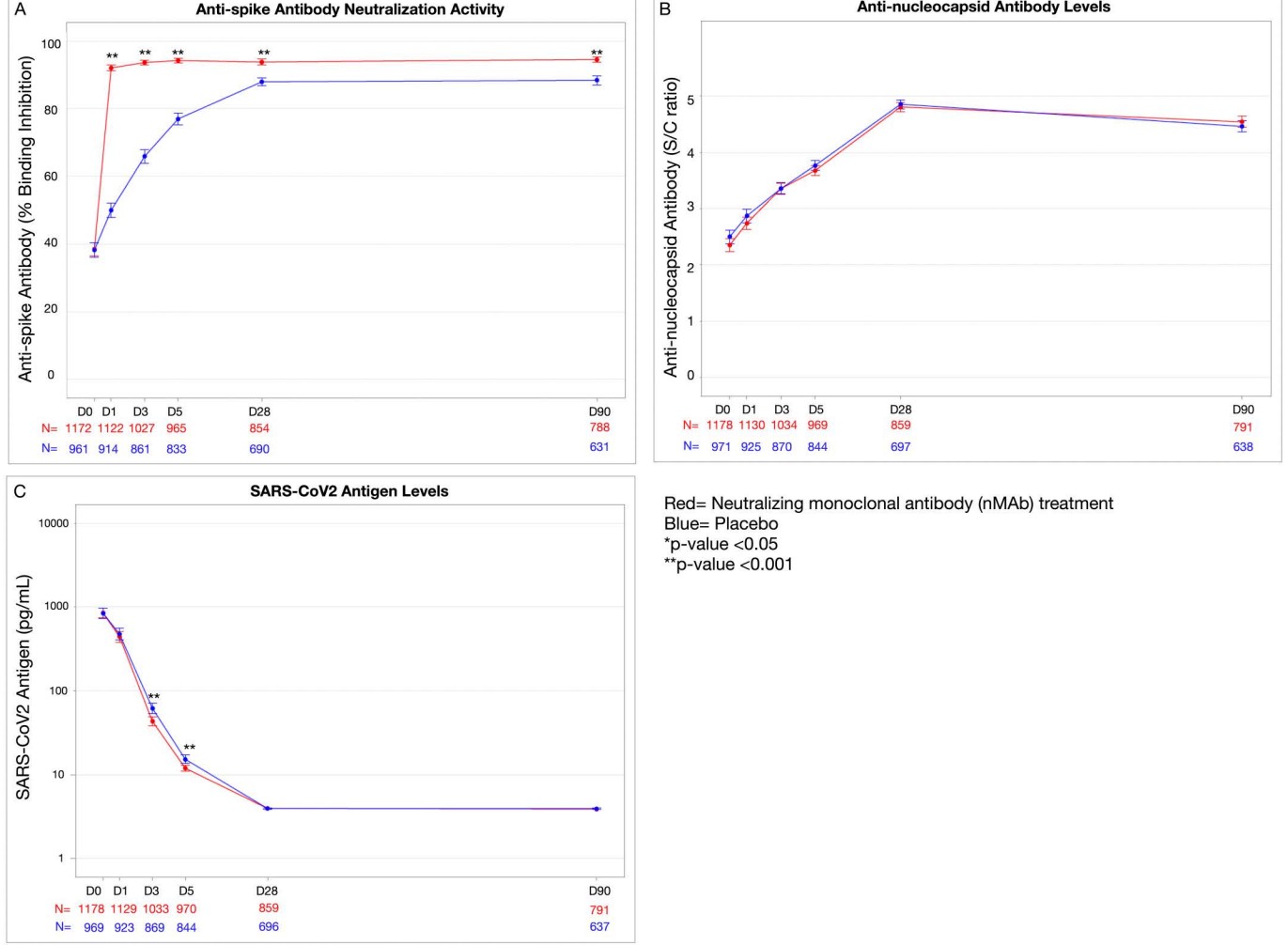

**Fig 2. Antibody and antigen responses after treatment with nMAb vs placebo, day 1-90 (pooled nMAb treatments).** Pooled day 1-90 antibody and antigen responses for randomized controlled trials comparing treatment to placebo, where treatments included: Bamlanivimab (LY-CoV555, Eli Lilly and Company), Sotrovimab (VIR-7831, Vir biotechnology and GlaxoSmithKline), Amubarvimab/ Romlusevimab (BRII-196 and BRII-198, BRII Biosciences), and Tixagevimab/ Cilgavimab (AZD7442, AstraZeneca). **Panel A**: Anti-SARS-CoV-2 spike protein neutralization activity presented as percent binding inhibition (GenScript, Piscataway, New Jersey), **Panel B**: Total immunoglobulin (IgG, IgA, and IgM) against the SARS-CoV-2 nucleocapsid antigen presented as signal to cut off (S/C) ratio (BioRad, Hercules, California), **Panel C**: SARS-CoV-2 nucleocapsid antigen levels presented as pg/mL on a log scale (Quanterix, Billerica, MA). Values in plot are mean +/- standard error. P-values are from a longitudinal model comparing treatment versus placebo groups at each time point, adjusting for baseline level of blood marker. * = p-value < 0.05; ** = p-value < 0.001.

complex infections, such as malaria [43]. Antibody-mimicking proteins, such as the DARPin ensovibep, have similar neutralizing effects against viruses but have simpler structures that make them even easier to produce than nMAbs [44]. However, there is a need to understand the effect of these neutralizing therapies on the endogenous immune response [22,45].

Several prior studies have examined the effect of nMAbs on the long-term endogenous immune response to COVID-19 in the outpatient setting, finding only small effects [18]. An unplanned *post-hoc* analysis of patients in the BLAZE-2 phase III trial, which evaluated the preventative use of Bamlanivimab in nursing home residents and staff, showed no effect of

nMAb treatment on the endogenous immune response to subsequent COVID-19 vaccination [23]. In contrast, an earlier *post-hoc* analysis of the BLAZE-1 trial in outpatients with mild-to-moderate COVID-19 found two- to three-fold reductions in day 15–85 anti-nucleocapsid antibody titers and neutralizing activity after treatment with Bamlanivimab (with or without Etesevimab) [19]. Similarly, a small retrospective study of high risk outpatients with mild-to-moderate COVID-19 infection who received Bamlanivimab (with or without Etesevimab) or Casirivimab with Imdevimab found that day 20–40 endogenous anti-spike IgM levels were reduced in all patients treated with nMAbs compared to untreated patients and that the anti-nucleocapsid response was significantly reduced (up to 50%) in patients treated with Casirivimab/Imdevimab but not Bamlanivimab [21].

This study builds on these outpatient studies, finding that in patients hospitalized with COVID-19, treatment with SARS-CoV-2 nMAbs did not appear to suppress anti-nucleocapsid antibody levels. Long-term anti-spike neutralization activity was higher in patients treated with nMAbs. Given the long half-life of some nMAbs, particularly Amubarvimab/Romlusevimab (90 days) [10] and Tixagevimab/Cilgavimab (87.9/ 82.9 days) [46,47] [Table 1], these long-term differences in anti-spike neutralization activity may represent a combination of both treatment and endogenous response. In contrast, given the nucleocapsid protein was not targeted by any studied nMAb, it can be used as a surrogate for the endogenous immune response. We found no differences in short or long-term anti-nucleocapsid response between patients in the treatment vs placebo groups in pooled analysis and for each studied nMAb or combination of nMAbs. It is important to note that the clinical significance of small changes in antibodies is unknown. While nucleocapsid antibodies are not a demonstrated mechanistic correlate of protection, these responses likely track with the larger polyclonal B cell response to SARS-CoV-2 infection, which does provide immunity against re-infection, hospitalization and severe disease [48,49].

Compared to earlier studies, the ACTIV-3/TICO trials enrolled hospitalized patients who were likely further into their infectious course than the ambulatory patients with mild-to-moderate COVID-19 who were enrolled in other trials. One would expect that treating patients with neutralizing therapies later in their infectious course may have less impact on the host humoral immune response than earlier treatment, as patients have had time to mount a humoral immune response prior to treatment. Yet, only 50−60% of patients in the ACTIV-3/TICO trials met the threshold for positive anti-spike neutralizing activity (>30% binding inhibition) at baseline, highlighting the importance of understanding the effect of nMAbs on the humoral response even in hospitalized patients [Table 3].

While we did not observe a long-term difference in anti-nucleocapsid response after treatment with nMAbs, we observed that patients who had positive anti-spike neutralization activity at baseline had higher long-term anti-nucleocapsid response regardless of treatment assignment. Initial differences in humoral immune response by baseline antibody positivity would be expected, given patients who are positive for anti-spike neutralizing activity at baseline may be further into their infectious course. However, differences in initial infectious time courses do not explain the sustained difference in anti-nucleocapsid response by baseline anti-spike positivity at days 28 and 90. This long-term differential anti-nucleocapsid response suggests that other patient factors may be mitigating the host humoral immune response. For example, we found that patients with baseline immunosuppression had lower baseline antibody positivity and lower 28 and 90-day anti-nucleocapsid response, suggesting these patients had a less robust humoral immune response [50]. Fully vaccinated patients also had lower 28 and 90-day anti-nucleocapsid levels, though this finding is difficult to interpret given the point in the pandemic where the ACTIV-3/TICO trials were performed—during this period, vaccination was often reserved for patients with immunocompromise and significant comorbidities leading to risk of confounding by indication for vaccination. Conversely, patients on oxygen at baseline had higher baseline and long-term anti-nucleocapsid response, suggesting patients who are sicker, perhaps with increased viral burden or a stronger innate inflammatory response, may mount a stronger long-term humoral immune response. However, these analyses were exploratory and were not performed with robust risk adjustment. Therefore, the findings could be the result of confounding and are hypothesis-generating only. Furthermore, the clinical significance of these small differences in antibody levels on infection response is unclear.

Our study has several strengths. The ACTIV-3/TICO trials took place at multiple sites around the world and spanned two SARS-CoV-2 variants (Ancestral and Delta). Unlike many prior studies, the analysis of long-term antibody responses was pre-specified in the ACTIV-3/TICO platform. Therefore, 28 and 90-day specimen collections were pre-planned and standardized. This allowed us to pool results across trials, yielding a large sample size. Additionally, while prior studies have focused on the effects of Bamlanivimab and Etesevimab, some of the earliest available nMAbs, this study examined the long-term antibody responses across multiple nMAbs and novel therapies, to assess for broader class effects.

Our study has several limitations. First, we used anti-nucleocapsid antibody levels as a surrogate for the endogenous immune response. While this approach has been used in prior COVID studies [18,19], anti-nucleocapsid levels represent only part of the endogenous immune response and may not correlate to clinical immune protection. Secondly, the therapies studied in ACTIV-3/TICO did not lead to improved clinical outcomes in hospitalized patients and are of limited clinical value going forward because of evolution of the SARS-CoV-2 virus [12–17,51]. However, our results provide insight into the long-term effects of neutralizing therapies, which remain an important tool for treating novel viral infections given the feasibility of rapid development and deployment. In addition, the patients included in this study were hospitalized and potentially already mounting an adaptive immune response. Therefore, we were not able to assess whether nMAbs suppress early immune responses and the generalizability of our findings to earlier treatment with neutralizing therapies, for prevention or in less severe disease, is not clear. Additionally, the studied nMAbs had variable pharmacokinetics, potentially complicating the interpretation of the pooled anti-spike analysis. For example, many of the newer nMAbs are engineered to prolong their circulation and several, including Amubarvimab/Romlusevimab and Tixagevimab/Cilgavimab, have half-lives of almost 90 days (Table 1), limiting the ability to distinguish the effect of treatment vs the innate anti-spike antibody response. However, the individual analyses of each studied nMAb/ nMAb combination supported the pooled results. Also, we were unable to evaluate the effect of recurrent COVID-19 infection on long-term antibody response, given data on recurrent infections were not collected in these trials. Finally, our study was limited by loss to follow-up, with approximately 70% of participants having day-90 samples, though results were robust to sensitivity analysis including only patients alive and with antibody measured at day 90.

In conclusion, in patients hospitalized with COVID-19, treatment with SARS-CoV-2 anti-spike nMAbs vs placebo led to similar 28 and 90-day anti-nucleocapsid response, suggesting these treatments did not suppress the endogenous humoral immune response. Because the SARS-CoV-2 virus has evolved, these nMAbs are no longer beneficial for preventing and treating COVID-19. However, these findings suggest that nMAbs and other targeted therapies like DARPins remain promising strategies for treating patients hospitalized with novel infections without impairing the host humoral immune response.

## Supporting information

**S1 Fig. Antibody and antigen responses to nMAb vs placebo day 1–90 for Bamlanivimab.** Antibody and antigen responses for the randomized controlled trial comparing treatment with the neutralizing monoclonal antibody Bamlanivimab (LILLY) vs placebo. **Panel A**: Anti-SARS-CoV-2 spike protein neutralization activity presented as percent binding inhibition (GenScript, Piscataway, New Jersey), **Panel B**: Total immunoglobulin (all immunoglobulin types) against the SARS-CoV-2 nucleocapsid antigen presented as signal-to-cutoff ratio (BioRad, Hercules, California), **Panel C**: SARS-CoV-2 nucleocapsid antigen levels presented as pg/mL on a log scale (Quanterix, Billerica, MA). * = p-value < 0.05; ** = p-value <0.001. (TIFF)

**S2 Fig. Antibody and antigen responses to nMAb vs placebo day 1–90 for Sotrovimab.** Antibody and antigen responses for the randomized controlled trial comparing treatment with the neutralizing monoclonal antibody Sotrovimab (VIR) vs placebo. **Panel A**: Anti-SARS-CoV-2 spike protein neutralization activity presented as percent binding inhibition (GenScript, Piscataway, New Jersey). Of note, Sotrovimab, which blocks viral fusion, recognizes a proteoglycan epitope

distinct from the receptor binding motif itself and therefore is not detected by this GenScript assay. **Panel B**: Total immunoglobulin (all immunoglobulin types) against the SARS-CoV-2 nucleocapsid antigen presented as signal-to-cutoff ratio (BioRad, Hercules, California). **Panel C**: SARS-CoV-2 nucleocapsid antigen levels presented as pg/mL on a log scale (Quanterix, Billerica, MA). * = p-value < 0.05; ** = p-value <0.001.
(TIFF)

**S3 Fig. Antibody and antigen responses to nMAb vs placebo day 1–90 for Amubarvimab/ Romlusevimab.** Antibody and antigen responses for the randomized controlled trial comparing treatment with the neutralizing monoclonal antibody Amubarvimab/ Romlusevimab (BRII) vs placebo. **Panel A**: Anti-SARS-CoV-2 spike protein neutralization activity presented as percent binding inhibition (GenScript, Piscataway, New Jersey), **Panel B**: Total immunoglobulin (all immunoglobulin types) against the SARS-CoV-2 nucleocapsid antigen presented as signal-to-cutoff ratio (BioRad, Hercules, California), **Panel C**: SARS-CoV-2 nucleocapsid antigen levels presented as pg/mL on a log scale (Quanterix, Billerica, MA). * = p-value < 0.05; ** = p-value <0.00.
(TIFF)

**S4 Fig. Antibody and antigen responses to nMAb vs placebo day 1–90 for Tixagevimab/ Cilgavimab.** Antibody and antigen responses for the randomized controlled trial comparing treatment with the neutralizing monoclonal antibody Tixagevimab/Cligavimab (AZ) vs placebo. **Panel A**: Anti-SARS-CoV-2 spike protein neutralization activity presented as percent binding inhibition (GenScript, Piscataway, New Jersey), **Panel B**: Total immunoglobulin (all immunoglobulin types) against the SARS-CoV-2 nucleocapsid antigen presented as signal-to-cutoff ratio (BioRad, Hercules, California), **Panel C**: SARS-CoV-2 nucleocapsid antigen levels presented as pg/mL on a log scale (Quanterix, Billerica, MA). * = p-value < 0.05; ** = p-value <0.001.
(TIFF)

**S5 Fig. Antibody and antigen responses to Ensovibep vs placebo day 1–90.** Antibody and antigen responses for the randomized controlled trial comparing treatment with ensovibep (MP) vs placebo. Ensovibep (MP) is a designed ankyrin repeat protein (DARPin) which targets and neutralizes the SARS-CoV-2 spike protein. **Panel A**: Anti-SARS-CoV-2 spike protein neutralization activity presented as percent binding inhibition (GenScript, Piscataway, New Jersey), **Panel B**: Total immunoglobulin (all immunoglobulin types) against the SARS-CoV-2 nucleocapsid antigen presented as signal-to-cutoff ratio (BioRad, Hercules, California), **Panel C**: SARS-CoV-2 nucleocapsid antigen levels presented as pg/mL on a log scale (Quanterix, Billerica, MA). * = p-value < 0.05; ** = p-value <0.001.
(TIFF)

**S6 Fig. Antibody and antigen responses to Lufotrelvir vs placebo day 1–90.** Antibody and antigen responses for the randomized controlled trial comparing treatment with lufotrelvir (PF) vs placebo. Lufotrelvir (PF) is a phosphate ester prodrug that is a selective inhibitor of the SARS-CoV-2 3CLpro, a viral proteinase. **Panel A**: Anti-SARS-CoV-2 spike protein neutralization activity presented as percent binding inhibition (GenScript, Piscataway, New Jersey). Of note, Lufotrelvir targets viral assembly and therefore is not detected by this assay. **Panel B**: Total immunoglobulin (all immunoglobulin types) against the SARS-CoV-2 nucleocapsid antigen presented as signal-to-cutoff ratio (BioRad, Hercules, California). **Panel C**: SARS-CoV-2 nucleocapsid antigen levels presented as pg/mL (Quanterix, Billerica, MA). P-values not presented given small sample size (N = 51 at day 0, N = 25 at day 90).
(TIFF)

**S7 Fig. Longitudinal antibody and antigen responses to treatment vs placebo day 1–90 for nMAbs, among patients alive and with day 90 antibody and antigen measures (Sensitivity Analysis).** Pooled antibody and antigen responses for randomized controlled trials comparing treatment with neutralizing monoclonal antibodies (nMAbs)

to placebo among patients alive and with all antibody/antigen measurements available at day 90. NMAbs included: Bamlanivimab (LILY), Sotrovimab (VIR), Amubarvimab/ Romlusevimab (BRII), Tixagevimab/ Cilgavimab (AZ). **Panel A**: Anti-SARS-CoV-2 spike protein neutralization activity presented as percent binding inhibition (GenScript, Piscataway, New Jersey), **Panel B**: Total immunoglobulin (all immunoglobulin types) against the SARS-CoV-2 nucleocapsid antigen presented as signal-to-cutoff ratio (BioRad, Hercules, California), **Panel C**: SARS-CoV-2 nucleocapsid antigen levels presented as pg/mL on a log scale (Quanterix, Billerica, MA). $*$ = p-value < 0.05; $**$ = p-value <0.001.
(TIFF)

**S1 Table. Longitudinal antibody and antigen responses to treatment vs placebo day 1–90 for neutralizing monoclonal antibodies, among patients alive and with day 90 antibody and antigen measures (Sensitivity Analysis).** Analysis using mixed model analysis (SAS Mixed procedure), adjusted for baseline antibody/ antigen values. At least one follow-up value required to be in analysis. Pooled Days 1–90 represents the treatment effect pooled over all follow-up days.
(TIFF)

**S8 Fig. Antibody and antigen responses by baseline anti-spike neutralizing response assay positivity.** Pooled antibody and antigen responses for randomized controlled trials comparing treatment with all ACTIV-3/TICO treatments (nMAbs, ensovibep, and lufotrelvir) to placebo, by positive baseline anti-spike antibody neutralization activity (>30% binding inhibition). Green and grey lines represent patients with negative anti-spike protein neutralization activity at baseline, while blue and red lines represent patients with positive anti-spike protein neutralization activity at baseline. **Panel A**: Anti-SARS-CoV-2 spike protein neutralization activity presented as percent binding inhibition (GenScript, Piscataway, New Jersey), **Panel B**: Total immunoglobulin (all immunoglobulin types) against the SARS-CoV-2 nucleocapsid antigen presented as signal to cut off ratio (BioRad, Hercules, California), **Panel C**: SARS-CoV-2 nucleocapsid antigen levels presented as pg/mL (Quanterix, Billerica, MA). P-values not displayed.
(TIFF)

**S9 Fig. Long-term anti-nucleocapsid response by baseline patient characteristics.** Pooled anti-nucleocapsid antibody levels for all included ACTIV-3/TICO trials based on patient subgroups. A) age (≥65 years old), B) sex, C) body mass index (BMI), D) history of diabetes, E) history of chronic kidney failure, F) immunosuppressed at baseline⊥, G) fully vaccinated was defined as patients who had completed a full course of COVID-19 vaccination and who had symptoms develop >14 days after last vaccine dose, H) baseline oxygen status (<4 Liters nasal cannula at trial enrollment vs ≥ 4 Liters nasal cannula). Anti-nucleocapsid antibody levels were measured using BioRad AB, which measures total immunoglobulin (all immunoglobulin types) against the SARS-CoV-2 nucleocapsid antigen (BioRad, Hercules, California). Results presented as signal-to-cutoff ratios. $*$ = p-value < 0.05; $**$ = p-value <0.001; ⊥Immunosuppression at baseline included patients with: an immunosuppressive disorder other than HIV, malignancy (active or receiving treatment), treatment with anti-rejection medications after solid or stem cell transplant, treatment with biological medicines used to treat autoimmune diseases or cancer, OR treatment with immune-modulators (e.g., interleukin-1 inhibitors, interleukin-6 inhibitors, TNF-inhibitors, etc).
(TIFF)

## Acknowledgments

The authors would like to thank the participants in the ACTIV-3/TICO trials and the study teams. See Supplement Acknowledgement file for a full list of STRIVE Network and Therapeutics for Inpatients with COVID-19 (TICO) study team members. This material is the result of work supported with resources and use of facilities at the Ann Arbor VA Medical Center. This manuscript reflects views of the authors and does not represent the views of the Department of Veterans Affairs or the US government. Data are posted to https://public-data.ccbr.umn.edu/.

## Author contributions

**Conceptualization:** Elizabeth S. Munroe, Greg A. Grandits, Robert C. Hyzy, Hallie C. Prescott, Thomas W. Barrett, Robin L. Dewar, Nicole Engen, Anna L. Goodman, Timothy J. Hatlen, Helene Highbarger, Thomas L. Holland, Gareth Hughes, Tomas O. Jensen, Muhammad A. Khan, Ioannis Kalomenidis, Nayon Kang, Sylvain Laverdure, Prasad Manian, Vidya Menon, Ravi Patel, Srikanth Ramachandruni, Tauseef Rehman, Kathryn Shaw-Saliba, Birgit Thorup Røge, David M. Vock, Amy C. Weintrob, Barnaby E. Young, Anne P. Frosch.

**Data curation:** Greg A. Grandits.

**Formal analysis:** Greg A. Grandits, David M. Vock, Anne P. Frosch.

**Investigation:** Elizabeth S. Munroe, Hallie C. Prescott, Thomas W. Barrett, Robin L. Dewar, Nicole Engen, Anna L. Goodman, Timothy J. Hatlen, Helene Highbarger, Thomas L. Holland, Gareth Hughes, Tomas O. Jensen, Muhammad A. Khan, Ioannis Kalomenidis, Nayon Kang, Sylvain Laverdure, Prasad Manian, Vidya Menon, Ravi Patel, Srikanth Ramachandruni, Tauseef Rehman, Kathryn Shaw-Saliba, Birgit Thorup Røge, David M. Vock, Amy C. Weintrob, Barnaby E. Young, Anne P. Frosch.

**Methodology:** Elizabeth S. Munroe, Greg A. Grandits, Robert C. Hyzy, Hallie C. Prescott, Robin L. Dewar, Anna L. Goodman, Timothy J. Hatlen, Helene Highbarger, Thomas L. Holland, Gareth Hughes, Tomas O. Jensen, Muhammad A. Khan, Ioannis Kalomenidis, Nayon Kang, Sylvain Laverdure, Prasad Manian, Vidya Menon, Ravi Patel, Tauseef Rehman, Kathryn Shaw-Saliba, Birgit Thorup Røge, David M. Vock, Amy C. Weintrob, Barnaby E. Young, Anne P. Frosch.

**Project administration:** Elizabeth S. Munroe, Greg A. Grandits, Anne P. Frosch.

**Supervision:** Robert C. Hyzy, Hallie C. Prescott, Anne P. Frosch.

**Validation:** Elizabeth S. Munroe, Greg A. Grandits, Robert C. Hyzy, Hallie C. Prescott, Robin L. Dewar, Tomas O. Jensen, Kathryn Shaw-Saliba, David M. Vock, Anne P. Frosch.

**Visualization:** Elizabeth S. Munroe, Greg A. Grandits, Anne P. Frosch.

**Writing – original draft:** Elizabeth S. Munroe, Greg A. Grandits, Hallie C. Prescott, Anne P. Frosch.

**Writing – review & editing:** Greg A. Grandits, Robert C. Hyzy, Hallie C. Prescott, Thomas W. Barrett, Robin L. Dewar, Nicole Engen, Anna L. Goodman, Timothy J. Hatlen, Helene Highbarger, Thomas L. Holland, Gareth Hughes, Tomas O. Jensen, Muhammad A. Khan, Ioannis Kalomenidis, Nayon Kang, Sylvain Laverdure, Prasad Manian, Vidya Menon, Ravi Patel, Srikanth Ramachandruni, Tauseef Rehman, Kathryn Shaw-Saliba, Birgit Thorup Røge, David M. Vock, Amy C. Weintrob, Barnaby E. Young, Anne P. Frosch.

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
