## [Decision Letter · Decision Letter 0]

Dear Dr. Munroe,

Thank you for submitting your manuscript to PLOS ONE. After careful consideration, we feel that it has merit but does not fully meet PLOS ONE’s publication criteria as it currently stands. Therefore, we invite you to submit a revised version of the manuscript that addresses the points raised during the review process.

The study aimed to clear an issue raised using nMAbs, and the findings the authors obtained might help support the planning of further studies.

Some considerations and suggestions follow below.

Even though the authors have pointed out that their current study is “a secondary analysis of trials conducted on the ACTIV-3/TICO platform”, I suggest including the approval number and ID given by the Ethical Committee in the manuscript.

142-145 line: could the authors go into the specifics of what trials they have been referring to with “from two other ACTIV-3/TICO placebo-controlled trials”?

146 line: Could they name the small molecule viral proteinase they cited in the text?

I recommend exploring the clinical features of hospitalized patients in more depth throughout the manuscript and expanding the description of the results. The discussion is well-articulated, but it also highlights the study's limitations. Therefore, I suggest the authors emphasize the findings' relevance more thoroughly.

We look forward to receiving your revised manuscript.

Kind regards,

Elisabetta Pilotti

Academic Editor

PLOS ONE

 [This research was, in part, funded by the National Institutes of Health (NIH) Agreement 1OT2HL156812-01. The trial was sponsored and primarily funded by the National Institute of Allergy and Infectious Diseases (NIAID), National Institutes of Health (NIH), Bethesda, MD, in part with federal funds from the NIAID and the National Cancer Institute, NIH, under contract 75N91019D00024, task order numbers 75N91020F00014 and 75N91020F00039.

The work was funded under Subcontract 18X107C under Leidos Biomeds's Prime Contract HHSN261200800001E, NIH. NIH Grant U01-AI136780.

The following authors received relevant funding support:

•            Author ESM was supported by Grant Number F32 HL 172463 from the National Institutes of Health, National Heart, Lung, and Blood institute.

•            Author BEY was supported by Singapore National Medical Research Council (NMRC, grant number COVID19RF-0005).

•            Author ALG receives funding to support salary from the Medical Research Council (MC_UU_00004/05) ]. 

5. For studies involving third-party data, we encourage authors to share any data specific to their analyses that they can legally distribute. PLOS recognizes, however, that authors may be using third-party data they do not have the rights to share. When third-party data cannot be publicly shared, authors must provide all information necessary for interested researchers to apply to gain access to the data. (https://journals.plos.org/plosone/s/data-availability#loc-acceptable-data-access-restrictions)

4) All necessary contact information others would need to apply to gain access to the data.

6. One of the noted authors is a group [the STRIVE Network and Therapeutics for Inpatients with COVID-19 (TICO) study groups.]. In addition to naming the author group, please list the individual authors and affiliations within this group in the acknowledgments section of your manuscript. Please also indicate clearly a lead author for this group along with a contact email address.

Reviewers' comments:

Reviewer's Responses to Questions

**Comments to the Author**

1. Is the manuscript technically sound, and do the data support the conclusions?

Reviewer #1: No

Reviewer #2: Yes

2. Has the statistical analysis been performed appropriately and rigorously?

Reviewer #1: No

Reviewer #2: Yes

3. Have the authors made all data underlying the findings in their manuscript fully available?

Reviewer #1: No

Reviewer #2: Yes

4. Is the manuscript presented in an intelligible fashion and written in standard English?

Reviewer #1: Yes

Reviewer #2: Yes

Reviewer #1: The study design does not fully align with the stated objective. While the manuscript claims to assess whether nMAbs impair the endogenous humoral response, the approach using anti-nucleocapsid antibody levels as a proxy is not sufficiently justified. Since nMAbs target spike proteins, they are not expected to affect nucleocapsid-specific responses, making the conclusion potentially misleading.

The heterogeneity in the half-lives of different nMAbs used (ranging from 20 to 90 days) further complicates interpretation. Pooling all nMAbs into a single treatment group without addressing this variability weakens the validity of the conclusions.

The conclusion that “nMAbs did not impair endogenous immune response” is not directly supported by the study design, as the selected immune markers do not sufficiently capture the full scope of humoral immunity.

The manuscript lacks mechanistic insight into how neutralizing monoclonal antibodies could potentially influence long-term immune responses, making the proposed hypothesis difficult to evaluate within the current study framework.

The manuscript does not sufficiently account for potential confounding factors, including reinfections or differences in disease severity, which could influence long-term antibody responses.

The choice of a placebo as a control in evaluating the impact of nMAbs on endogenous immune responses is questionable. Since nMAbs were administered to patients at different points in their infection course, a before-and-after study design where each patient serves as their own control might have been a more appropriate approach.

The variability in nMAb pharmacokinetics is not addressed in the statistical analysis, which could lead to misleading interpretations.

There is no clear indication that data has been deposited in a public repository or made available as supporting information. Given that this study is a secondary analysis of a clinical trial, it is important to clarify whether the dataset can be shared, and if not, what restrictions apply.

Reviewer #2: This is a nice analysis drawing on data from a clinical trials platform. A few minor comments:

Lines 131-132: “In SARS-CoV-2 in particular, previous small studies have demonstrated decreases in IgM, IgG and neutralizing activity with monoclonal antibody treatments” – it would be helpful to clarify what type of IgM, IgG, and neutralizing responses.

Line 141 and Figure 1: Would use “participants” instead of “patients”

In multiple places, authors refer to four nMAbs and “individual agents”, but it’s really four trials of a total of 6 mAbs, since two of the trials looked at a combination of two mAbs. Would clarify throughout.

Lines 186-187: This could be clarified. It seems authors mean a participant had to have a measurement at day 1, 3, or 5 to be included in the mITT analysis, but it’s not completely clear as written.

Lines 195-201: Authors discuss at various points that the antibody levels are measured, but it is the neutralization that is measured for anti-spike Ab, not the amount of immunoglobulin present. Throughout the manuscript, authors sometimes refer to “antibodies at baseline”. Language should be clarified. Also applies to Figure 2 and many of the supplementary figures.

Line 222 – should be each mAb or each mAb combination, since for two of the studies that combined mAbs, those mAbs could not be measured alone

Line 325: Given recent promising results in malaria vaccine studies, would not characterize malaria as an infection for which vaccines have failed

Lines 327-330 are covered earlier in intro section and could be cut.

Lines 333-344: It might be helpful to give more context/details around these earlier studies. Were the changes statistically significant? What was the magnitude of the changes?

A number of typos throughout, e.g., punctuation on line 226, “measures” rather than “measured” on Line 187, missing word on Line 230, Line 140, and others.

**Do you want your identity to be public for this peer review?** For information about this choice, including consent withdrawal, please see our Privacy Policy

Reviewer #1: No

Reviewer #2: No

---

## [Author Response · Author response to Decision Letter 1]

10 May 2025

Editor Comments

I appreciate the effort invested in conducting this study and recognise its significance in addressing concerns related to the use of neutralising monoclonal antibodies (nMAbs) in hospitalized COVID-19 patients. The study leverages a large dataset from the ACTIV-3/TICO platform and aims to assess the impact of nMAbs on endogenous humoral immune responses. Given the importance of understanding the potential immunological consequences of nMAb treatment, this study has the potential to contribute valuable insights to the field. However, several critical issues need to be addressed to ensure the manuscript accurately represents the findings and effectively answers the research question.

Thank you for your feedback and the opportunity to address these concerns.

1. Study Design and Alignment with Study Objective

1. The study is a secondary analysis of a treatment trial for nMAbs, yet the current framing implies it is a primary investigation into endogenous immune responses.

R1.1: We had divided our analysis into primary (pooled trials) and secondary (individual trials) analyses based on our pre-specified analysis plan. However, as this comment highlights, both of these analyses are secondary analyses of the ACTIV-3/TICO trials. Therefore, in the Abstract and Methods section, we have clarified that our analysis is a secondary analysis of the ACTIV-3/TICO trials that examined the secondary endpoint of antibody response on this trial platform. We have also changed the language to avoid calling these pooled vs individual analyses primary and secondary to avoid confusion (Methods, line 209).

2. The study design is not adequately designed to address the stated objective of evaluating whether nMAbs impair long-term immune response.

R1.2: We agree that our study only evaluated one antigen target of the endogenous humoral immune response. All studies on natural immunity necessarily choose markers of a larger polyclonal response. We used anti-nucleocapsid response as a surrogate for the endogenous immune response because it was a standardized assay that was widely available in clinical practice as a marker of natural SARS CoV-2 exposure at the time of study design. This approach is consistent with prior studies of anti-SARS-CoV2 nMAbs, as we now highlight in the Introduction (lines 112-120). However, we agree that nucleocapsid response is not a mechanistic correlate of protection from future infection or disease. We have adjusted language throughout the manuscript to specify that we measured anti-nucleocapsid response and to make it clear that while we are using nucleocapsid response as a proxy for endogenous response, the clinical correlation between anti-nucleocapsid levels and immune protection is unknown. We have also specifically added this as a limitation in the Discussion (lines 401-404).

3. The choice of anti-nucleocapsid antibodies as a proxy for endogenous humoral response to SARS-CoV-2 is questionable. Since the nMAbs studied target the spike protein, there is no mechanistic rationale to expect any impact on nucleocapsid-specific responses. Why was this measurement was used to infer immune suppression effects of nMAbs.

R1.3: The use of anti-nucleocapsid antibodies as a proxy for endogenous humoral response in COVID has been well established in the literature (PMID: 38013970, PMID: 34956222). The mechanistic rationale by which a monoclonal Ab therapy might impair the endogenous polyclonal immune response is that neutralization of the SARS-CoV-2 virus through anti-spike targeted monoclonal antibodies could result in decreased presentation of all viral antigens to B cells and thus lead to a decreased host response to the virus. Treatment with monoclonal Ab therapy could also modify the innate immune response by limiting viral replication, thereby altering the humoral immune response to all viral antigens. While this biology is not fully understood in natural SARS-CoV-2 immunity, both antigen dose and innate signaling are well characterized factors in the strength of the humoral immune response. Because anti-nucleocapsid antibodies are not directly affected by the anti-spike nMAb treatment, measuring anti-nucleocapsid antibody levels can serve as a proxy for the endogenous humoral response. Our study builds on prior studies that have used similar approaches in patients with mild to moderate COVID-19 infection who were treated with anti-spike nMAbs (PMID: 34956222). We have clarified the proposed mechanistic rationale for the potential negative impact of anti-spike nMAb treatment on the host humoral response and the rationale for using anti-nucleocapsid levels as a proxy for endogenous humoral response in the Introduction (lines 112-120) and comment on the limitations of this approach in the Discussion (lines 401-404).

2. Abstract Accuracy and Clarity

1. The abstract suggests that all randomized participants were tested, which is misleading. In the results section, it is stated that some participants died before follow-up. This needs to be addressed for clarity and accuracy.

R2.1: Thank you. We have updated the abstract to report the number of patients with plasma specimens available at Day 28 and Day 90.

2. The term "Day x interaction" in the results section is unclear and does not provide meaningful information. A more precise statistical description should be used.

R2.2: We have removed this wording from the Abstract and use a more precise statistical description in the Methods (line 217) and Results sections of the full text (line 269).

3. The conclusion in the abstract states that nMAbs did not impair endogenous humoral immune response, which does not align with the study design. Since nMAbs were anti-spike, they were not expected to affect nucleocapsid-specific responses, making the conclusion misleading. This needs to be rephrased to avoid overstating the findings.

R2.3: As discussed in R1.3 above, the concern is that even though nMAbs target the spike protein, by neutralizing virus activity treatment with nMAbs could suppress viral presentation on B-cells and thus impair the endogenous immune response. Because anti-nucleocapsid antibodies are not directly affected by the anti-spike nMAb treatment, the anti-nucleocapsid response can be used as a surrogate for the endogenous immune response (PMID: 38013970, PMID: 34956222). We have expanded the Introduction of both the Abstract and text (Lines 112-120) to outline the mechanistic rationale for measuring anti-nucleocapsid response. To avoid overstating findings, we have ensured that we consistently state that the lack of a difference in nucleocapsid response between treatment vs placebo groups only suggests that nMAbs do not affect endogenous immune response, though this cannot be measured directly. We also now comment on the limitations of this approach in the Discussion (Lines 401-404).

3. Presentation of Study Groups and Data Interpretation

1. The study groups would be better visualized using a flow diagram, the current description makes it difficult to follow how participants were selected, randomized, and analyzed.

R3.1 We have clarified the inclusion criteria and specimen availability in the Results section (line 242). We have also included a flow diagram (Fig 1).

2. Lines 169-170 mention that the different mAbs studied have varying half-lives ranging from 20 to 90 days. Thus, bundling these under a single treatment group introduces heterogeneity, making it difficult to interpret long-term immune response comparisons. Consider analyzing them separately.

R3.2. In addition to the combined group we have analyzed each nMAb trial separately. These results are presented in supplemental figures (S1-S4 Figs).

3. The placebo group's relevance in the analysis is unclear. If the objective is to assess changes in endogenous immune response due to nMAb treatment, a before-and-after study where each participant serves as their own control might have been more appropriate. The rationale for using a placebo should be explicitly stated.

R3.3. Most people infected by SARS-CoV-2 generate a polyclonal immune response to the virus that includes antibodies to both the spike protein and the nucleocapsid antigen. We present the “before” (baseline) results in Table 3 and Figure 2. However, while a before-and-after analysis of antibody activity/levels would be able to tell us if patients experienced a detectable response to treatment based on validated thresholds for antibody positivity, a placebo arm is needed to determine if that antibody response is lower or higher than would be expected without monoclonal antibody treatments. We have added a rationale for using a placebo group to the description of the Aims of the study in the Introduction (Line 127) “The aim of the study was to compare 28-day and 90-day responses with and without nMAb treatment.” We also clarified that the goal of the analysis was to compare antibody response between treatment vs placebo in the Methods (Line 139).

4. Statistical and Methodological Concerns

1. It is unclear whether there was any testing for reinfection during follow-up. If new infections occurred post-treatment, they could confound results related to endogenous immune response.

R4.1. This is an important point. However, re-infection was not evaluated in the ACTIV-3/TICO trials. We have added this as a limitation in the Discussion section.

2. In lines 271 onwards, it is expected that antibody levels would rise in the nMAb group since participants were infused with these antibodies. The novelty of reporting this as a study finding is unclear, the study should focus more on comparing endogenous responses.

R4.2: Yes, the rise in anti-spike antibody levels in the nMAb response is expected due to treatment. We explain this in the Results (Line 269): “Compared to placebo, anti-spike neutralization activity rose faster in patients who received nMAbs (p<0.001), reflecting treatment with the anti-spike nMAb.” We agree that this is not the focus of the manuscript. However, we think this information is important to report to help readers interpret the anti-spike results in the context of nMAb treatment. Further, there is some value in presenting the data on the 90-day differences in anti-spike neutralizing activity for non-monoclonal antibody agents (ensovibep and lufotrelvir), although we did not want to focus on these findings in the results or discussion given the small sample sizes. These agents each have shorter half-lives than the studied nMAbs and therefore are not likely present out to Day 90. We did not detect significant differences in the spike neutralization assay (Genscript) at Day 90 for these two agents (S Figure 5 and 6), supporting that the higher long-term anti-spike neutralization activity we observed in the nMAb treatment group likely reflects a combination of the effect of treatment and the endogenous response.

3. Lines 284-286 suggest that anti-spike antibodies reduced nucleocapsid antigen levels, but anti-spike antibodies are not expected to have any direct effect on nucleocapsid antigen clearance. This requires further explanation or reconsideration.

R4.3: These lines report the results of the anti-nucleocapsid response among patients in the treatment and placebo groups. No difference in response between the groups was observed. However, as discussed in the response to comment 1.3 above, there is a theoretical concern that treatment with monoclonal antibodies against the spike protein might neutralize the virus and prevent viral presentation by B-cells or alter innate immune signaling, thus blunting an anti-nucleocapsid response. Indeed, this was seen in a secondary analysis of the BLAZE-1 trial where outpatients with mild to moderate COVID-19 infection treated with nMAb had lower anti-nucleocapsid antibody levels than those treated with placebo (PMID: 34956222). Therefore, the finding presented in these lines is unique and important to report. In the Discussion (Line 362), we explain why we might have observed no difference in anti-nucleocapsid response with nMAb treatment while other studies, like the BLAZE-1 trial, observed differences, namely the focus on different patient populations.

5. Interpretation of Results and Conclusions

1. The statement that nMAbs "did not impair host response to the target antigen or other viral antigens" (line 318) is not supported by the study design. Since nucleocapsid was not a target of the nMAbs, its response cannot serve as an indicator of whether nMAbs impaired endogenous immunity.

R5.1: Our analysis aimed to address the concern that suppressing viral presentation to B cells by neutralizing the virus with nMAbs might impair the host response to not just the nMAb target antigen (spike protein) but also to other viral antigens. The finding of similar anti-nucleocapsid levels at days 28 and 90 between patients treated with nMAb vs placebo suggest treatment with nMAbs did not impair the host response to other non-target viral antigens (nucleocapsid). While these responses likely track with the larger polyclonal B cell response to SARS-CoV-2 (see Discussion line 357-360), we acknowledge that these measures are only surrogates for the host response and may not correlate to protection. Therefore, we have added this as a limitation in the Discussion (line 401-404)

2. The discussion should explicitly acknowledge that hospitalized patients may already be mounting an adaptive immune response by the time they receive nMAbs, making it difficult to assess whether nMAbs suppress early immune responses.

R5.2 Yes, we highlight this in the limitations section of the Discussion on line 408: “…the patients included in this study were hospitalized and potentially already mounting an adaptive immune response. Therefore, we were not able to assess whether nMAbs suppress early immune responses and the generalizability of our findings to earlier treatment with neutralizing therapies, for prevention or in less severe disease, is not clear.”

3. The exploratory epidemiologic analysis results need more careful interpretation. The observed differences in long-term anti-nucleocapsid responses by baseline antibody positivity, immunosuppression, and oxygen status may be influenced by confounding factors. These findings should not be overstated.

R5.3. Thank you. We agree. This analysis was hypothesis-generating only and must be interpreted with caution. Therefore, have added further discussion about how these findings may be due to confounding and are of unclear clinical significance in the Discussion (line 388): “However, these analyses were exploratory and the findings could be the result of confounding. Furthermore, the clinical significance of these small differences in antibody levels on infection response are unclear.”

6. Structural and Formatting Issues

1. Lines 254-269 primarily describe the study population and should be moved to the Methods section rather than appearing in the Results section.

R6.1: We have edited the referenced lines in response to other comments, to better explain the study flow and available specimens at day 28 and day 90. We have left this section in the Results (line 242-252) because the updated text now reports the results of who was included in the study. However, we would be happy to move this section to the Methods if that would be more appropriate.

2. The limitations section should acknowledge the use of nucleocapsid antibodies as a surrogate for endogenous response to anti-spike antibodies.

R6.2: Thank you. We agree that this is an important limitation. We have added this to the limitations section of the Discussion (lines 401-404).

Reviewer #1

The study design does not fully align with the stated objective. While the manuscript claims to assess whether nMAbs impair the endogenous humoral response, the approach using anti-nucleocapsid antibody levels as a proxy is not sufficiently justified. Since nMAbs target spike proteins, they are not expected to affect nucleocapsid-specific responses, making the conclusion potentially misleading.

R1.1. The use of anti-nucleocapsid antibodies as a proxy for endogenous humoral response in COVID has been well established in the literature (PMID: 38013970, PMID: 34956222). The mechanistic rationale by whic

---

## [Editor Report · Decision Letter 1]

Long-term anti-SARS-CoV-2 antibody trajectories after neutralizing monoclonal antibody treatment

PONE-D-24-54292R1

Dear Dr. Munroe,

We’re pleased to inform you that your manuscript has been judged scientifically suitable for publication and will be formally accepted for publication once it meets all outstanding technical requirements.

Kind regards,

Elisabetta Pilotti

Academic Editor

PLOS ONE

Additional Editor Comments (optional):

Even though the authors have pointed out that their current study is “a secondary analysis of trials conducted on the ACTIV-3/TICO platform”, I suggest including the approval data and ID given by the Ethical Committee in the manuscript.
---

## [Editor Report · Acceptance letter]

PONE-D-24-54292R1

PLOS ONE

Dear Dr. Munroe,

I'm pleased to inform you that your manuscript has been deemed suitable for publication in PLOS ONE. Congratulations! Your manuscript is now being handed over to our production team.

Kind regards,

on behalf of

Dr. Elisabetta Pilotti

Academic Editor

PLOS ONE